# Prediction of Neonatal Length of Stay in High-Risk Pregnancies Using Regression-Based Machine Learning on Computerized Cardiotocography Data

**DOI:** 10.3390/diagnostics15232964

**Published:** 2025-11-22

**Authors:** Bianca Mihaela Danciu, Maria-Elisabeta Șișială, Andreea-Ioana Dumitru, Anca Angela Simionescu, Bogdan Sebacher

**Affiliations:** 1Department of Obstetrics and Gynecology, Carol Davila University of Medicine and Pharmacy, 050474 Bucharest, Romania; bianca-mihaela.danciu@drd.umfcd.ro (B.M.D.); anca.simionescu@umfcd.ro (A.A.S.); 2Department of Obstetrics, Gynecology and Neonatology, National Institute for Maternal and Child Health “Alfred Rusescu”-Polizu, 127715 Bucharest, Romania; 3Department of Applied Informatics, Military Technical Academy “Ferdinand I” of Bucharest, 050141 Bucharest, Romania; maria.sisiala@mta.ro (M.-E.Ș.); ioana.dumitru@mta.ro (A.-I.D.); 4Department of Obstetrics and Gynecology, Filantropia Clinical Hospital, 050474 Bucharest, Romania

**Keywords:** computerized cardiotocography, neonatal length of stay, fetal growth restriction, gestational diabetes, gestational hypertension, cholestasis, machine learning, regression models

## Abstract

**Background/Objectives**: The management of high-risk pregnancies remains a major clinical challenge, particularly regarding the optimal timing of delivery, which has significant implications for both perinatal outcomes and healthcare costs. In this context, computerized cardiotocography (cCTG) offers an objective, non-invasive and cost-effective method for fetal surveillance, providing quantitative measures of heart rate dynamics that reflect autonomic regulation and oxygenation status. This study aimed to develop and validate regression-based machine learning models capable of predicting the duration of neonatal hospitalization—an objective and quantifiable indicator of neonatal well-being—using cCTG parameters obtained outside of labor, binary clinical variables describing the presence or absence of pregnancy pathologies, and gestational age at monitoring and at delivery. **Methods**: A total of 694 singleton high-risk pregnancies complicated by gestational diabetes, preexisting diabetes, intrahepatic cholestasis of pregnancy, pregnancy-induced or preexisting hypertension, or fetal growth restriction were enrolled. Twenty clinically relevant features derived from cCTG recordings and perinatal data were used to train and evaluate four regression algorithms: Random Forest, CatBoost, XGBoost, and LightGBM against a linear regression model with Ridge regularization serving as a benchmark. **Results**: Random Forest achieved the highest generalization performance (test *R*^2^ = 0.8226; RMSE = 3.41 days; MAE = 2.02 days), outperforming CatBoost (*R*^2^ = 0.7059), XGBoost (*R*^2^ = 0.6911), LightGBM (*R*^2^ = 0.6851) and the linear regression benchmark with Ridge regularization (*R*^2^ = 0.5699) while showing a consistent train–validation–test profile (0.9428 → 0.8042 → 0.8226). The error magnitude (≈2 days on average) is clinically interpretable for neonatal resource planning, supporting the model’s practical utility. These findings justify selecting Random Forest as the final predictor and its integration into a clinician-facing application for real-time length-of-stay estimation. **Conclusions**: Machine learning models integrating cCTG features with maternal clinical factors can accurately predict neonatal hospitalization duration in pregnancies complicated by maternal or fetal disease. This approach provides a clinically interpretable and non-invasive decision support tool that may enhance delivery planning, optimize neonatal resource allocation, and improve perinatal care outcomes.

## 1. Introduction

Fetal hypoxia remains one of the most critical concerns in perinatal medicine, as it can lead to irreversible neurological and systemic damage. Consequently, identifying reliable tools to prevent its occurrence has become a central objective in obstetric care. Across European hospitals, the incidence of fetal hypoxia ranges from 0.06% to 2.8% [1]. Despite its relatively low frequency, the condition carries major clinical implications, with outcomes ranging from asphyxia and stillbirth to neonatal encephalopathy, long-term neurodevelopmental disability, and even neonatal death [2,3,4,5,6].

Cardiotocography (CTG) marked a major advance in fetal monitoring, evolving from George H. Bell’s 1938 recordings of fetal cardiac activity into the modern method that simultaneously assesses fetal heart rate and uterine contractions, and has been widely used for intrapartum surveillance since the late 1960s [7,8]. Despite its clinical value, CTG interpretation remains limited by subjectivity, low predictive value, and inter-observer variability, highlighting the need for standardized and objective approaches to assess fetal well-being [9,10,11].

Two main approaches have been proposed to address the lack of standardization and the absence of universal guidelines for assessing fetal well-being using CTG. The first relates to the subjective nature of conventional CTG interpretation, which depends on clinicians’ visual assessment and is therefore influenced by differences in training, experience, and interpretation. In contrast, computerized CTG (cCTG) employs quantitative parameters that markedly reduce this subjectivity. Evidence from randomized controlled trials and retrospective studies indicates that cCTG-based decision support systems can improve interpretative accuracy while also shortening the time required for clinical decision-making [9,10,12].

The second challenge relates to the timing of CTG and cCTG recordings. Most previous studies have been conducted during labor, when transient episodes of fetal hypoxia are almost inevitable due to uterine contractions that temporarily reduce placental perfusion [8,12]. Such physiological fluctuations can obscure baseline fetal heart rate patterns and confound interpretation. Outside of labor, however, fetal hypoxia should generally be absent—except in cases of chronic hypoxemia associated with conditions such as intrauterine growth restriction (IUGR). Under these relatively stable conditions, both CTG and cCTG parameters may provide more reliable and unbiased information regarding fetal well-being. In this context, it is important to note that key cCTG analysis parameters, including short-term variability (STV) and long-term variability (LTV), have been validated primarily for term pregnancies during labor, and their interpretation outside this context remains insufficiently established [13].

Furthermore, most previous studies have primarily focused on predicting fetal acidosis or related biochemical outcomes. In particular, Petrozziello et al. [14] employed deep learning (DL) methods based on multimodal convolutional neural networks to analyze a large dataset of over 35,000 intrapartum CTG recordings, automatically predicting cord blood acidemia (pH < 7.05) as an objective indicator of fetal hypoxemia. This approach demonstrated that data-driven neural architectures can identify complex physiological patterns directly from raw signals, outperforming conventional feature-based or visually interpreted analyses. While such objectives are valuable in research settings, they have limited relevance to routine obstetric practice, where these biochemical measures are seldom used. In clinical contexts, CTG is primarily employed to detect real-time patterns suggestive of fetal hypoxemia. The present study addresses this gap by analyzing comprehensive cCTG recordings obtained at defined gestational ages outside of labor, aiming to develop predictive models with direct clinical applicability.

In light of these considerations, the development of advanced decision support tools specifically tailored for pregnancies complicated by maternal or fetal pathology appears particularly warranted. Such cases represent some of the most complex challenges in perinatal medicine, as clinicians must continuously balance the risks of prematurity against the detrimental effects of ongoing maternal disease on both mother and fetus. Determining the optimal timing of delivery is especially critical, as early intervention may lead to prolonged neonatal intensive care, whereas delayed delivery increases the risk of maternal or fetal deterioration [15,16]. In this context, computational systems capable of processing and interpreting large volumes of cCTG data—beyond the capacity of human agreement and assessment—offer the potential to provide evidence-based insights that can meaningfully add parameters to clinical decision making [14,17,18].

The implementation of such tools is considered an urgent need, not as a substitute for clinical expertise but as a complementary resource to support and enhance medical judgment. Recent studies have shown that integrating Artificial Intelligence-based (AI-based) decision support into perinatal care can improve risk stratification and reduce interobserver variability in CTG interpretation [19,20].

In recent years, both conventional CTG and cCTG data have been extensively analyzed using machine learning (ML) and DL techniques to identify abnormal fetal patterns and predict adverse perinatal outcomes. Traditional algorithms such as Random Forests, Support Vector Machines (SVM), and artificial neural networks have shown strong performance in classifying cCTG signals and correlating them with neonatal risk profiles. A large-scale multicenter study [21] involving 22,522 deliveries across 14 hospitals demonstrated that a DL-based system could classify CTG recordings with high accuracy (AUC 0.862–0.895 on external validation), confirming the potential of DL models for standardized, large-scale fetal surveillance and early detection of fetal distress.

Beyond DL, a wide range of ML-driven approaches have been investigated. Studies combining multiple algorithms within ensemble or stacking frameworks achieved superior accuracy (up to 98.9%), outperforming individual classifiers such as Logistic Regression, Decision Trees, SVM, and K-Nearest Neighbor (KNN) [22,23,24]. The integration of Gradient Boosting, Random Forest, and Voting Classifiers further improved multiclass fetal health prediction, reaching accuracies close to 99% and supporting interpretable model outputs for greater clinical transparency [23]. Robust feature selection was also found to enhance model reliability: filtering-based criteria, including ANOVA, ROC-AUC, and correlation-based methods (Pearson, Spearman, Kendall) increased prediction accuracy up to 92%, with Spearman correlation yielding the best overall results [25].

Several works have also demonstrated the effectiveness of hybrid ensemble techniques. A Blender Model combining twelve ML algorithms via soft voting achieved an Area Under the Curve (AUC) of 0.988 with high recall and precision [26], while bagging approaches such as Bagged Flexible Discriminant Analysis (bagFDA) reached 93.44% accuracy in high-risk maternal cohorts [27]. Association-based and weighted learning methods have been explored to mitigate class imbalance and feature dependency effects, achieving accuracies of 83–84% [28] and F1 scores up to 97.85% with weighted Random Forests [29,30]. Specialized architectures, including the Extreme Learning Machine (ELM) combined with Principal Component Analysis (PCA) for dimensionality reduction, obtained accuracies around 84% [31], whereas comparative studies identified Random Forests and Bagging ensembles as consistently top-performing models for binary CTG classification [32,33].

These findings collectively demonstrate the strong predictive capacity of ML and DL frameworks for CTG interpretation and fetal health assessment. However, despite this progress, most prior research has focused on classification of fetal status rather than quantitative outcome estimation. Few studies have investigated regression-based modeling of neonatal length of stay using clinically interpretable cCTG-derived features—a gap the present study aims to address.

All of the studies mentioned above, integrating conventional CTG or cCTG data with ML techniques, have focused mainly on fetal health assessment, while none have specifically addressed the prediction of neonatal length of hospital stay (LoS). Accurate LoS prediction remains a major clinical and logistical challenge, especially in neonatal and obstetric care, where timely interventions and optimal resource allocation are essential. ML offers valuable opportunities to address this problem by integrating gestational, perinatal, and maternal parameters, as well as CTG-derived indices, thereby enhancing both predictive accuracy and clinical decision support. Ensemble and interpretable ML approaches further promote model transparency and trust in medical practice.

A hybrid approach termed Classifier Fusion-LoS has been proposed to predict neonatal LoS in NICUs, using a binary framework that classifies short versus prolonged stays through a Voting Classifier combining multiple algorithms. This method achieved up to 96% accuracy and showed utility for hospital resource planning; however, its limited outcome granularity restricts broader clinical applicability [34]. DL models have also been explored for time-dependent LoS estimation. Using multivariate time-series data from the MIMIC-III database, architectures such as Long Short-Term Memory (LSTM) and convolutional neural networks outperformed logistic regression, enabling dynamic and continuous prediction of remaining hospital stay [35].

Regression-based ML models have similarly demonstrated strong potential in neonatal LoS estimation. Random Forest, CatBoost, and Linear Regression achieved an *R*^2^ of 0.82 when applied to large perinatal datasets, revealing interpretable predictors such as birth weight and gestational age. However, when the problem was reframed as multiclass classification, model performance decreased, reflecting the inherent complexity of predicting multiple LoS categories [36].

In broader ICU applications, ML and DL models—including Random Forest, XGBoost, CatBoost, LSTM, BERT, and Temporal Fusion Transformer—have been employed to predict patient LoS from clinical and demographic data in the MIMIC-IV database. Among these, Random Forest reached 68% accuracy, while BERT achieved balanced metrics (80% accuracy, precision, recall, and F1-score), illustrating the benefit of temporal modeling and transformer architectures for LoS prediction [37].

These advances confirm that integrating maternal, fetal, and perinatal data enhances risk assessment and supports individualized care strategies in high-risk pregnancies [38]. Nevertheless, cCTG interpretation should complement, not replace, clinical evaluation, maternal history, ultrasonography, and other investigations to ensure comprehensive assessment [39,40].

This paper introduces the development of an AI–based software tool that employs ML techniques to analyze cCTG data obtained at defined gestational ages outside of labor in high-risk pregnancies. The system is designed to estimate the expected duration of neonatal hospitalization according to the gestational age at delivery, thereby providing an objective framework to support clinical decision-making in high-risk pregnancies.

Four regression-based ML models—Random Forest, CatBoost, XGBoost, and LightGBM—were comparatively evaluated, with linear regression using Ridge regularization (implemented in scikit-learn, version 1.4.1.post1) included as a benchmark, to determine which algorithm provides the most accurate estimation of neonatal hospitalization duration. This comparative approach enabled the identification of the model with the best generalization performance and clinical interpretability for potential integration into a practical decision support tool. Similar predictive models using ML have been shown to improve the estimation of neonatal outcomes such as birth weight, preterm birth risk, and NICU admission duration [30,34,41].

Length of neonatal stay was selected as the primary outcome measure, as it represents one of the most objective and quantifiable indirect indicators of neonatal health status [42]. The dataset used for model development included cCTG recordings from pregnancies complicated by the most frequently encountered maternal conditions, gestational diabetes, intrahepatic cholestasis of pregnancy, and pregnancy-induced hypertension, as well as fetal growth restriction, which represents a distinctly fetal rather than maternal pathology. By incorporating these high-prevalence conditions, the model was designed to reflect clinically relevant scenarios and provide meaningful insights for routine obstetric care.

The proposed approach provides several advantages, including improved accuracy, efficiency, and interpretability, thereby supporting a more precise and accessible system for fetal health assessment with the potential to substantially improve maternal and neonatal outcomes. The method is entirely non-invasive and poses no risk to either mother or fetus, enabling objective and reproducible evaluation of fetal well-being throughout pregnancy, both before and during labor [1,17]. It facilitates the early identification of patients at risk of developing pathological conditions, even in the absence of clinical symptoms, and allows both the monitoring of disease progression and the evaluation of treatment effectiveness [17,19].

Beyond these clinical benefits, the method is also cost-effective. cCTG equipment is considerably less expensive than ultrasound devices and requires fewer human resources, as monitoring can be performed by midwives or nurses without advanced subspecialty training, whereas ultrasound assessment typically demands expertise in maternal–fetal medicine [43,44]. Collectively, these features highlight the potential of cCTG analysis and ML–based modeling as promising tools for improving perinatal outcomes in high-risk pregnancies.

## 2. Materials and Methods

### 2.1. Dataset

This retrospective observational study was conducted at the Filantropia Clinical Hospital of Obstetrics and Gynecology, Bucharest, Romania, between 2022 and 2025, as part of the doctoral research work of B.M.D. A total of 694 pregnant women were included. Eligibility criteria comprised singleton pregnancies with a gestational age of at least 28 weeks at the time of cCTG monitoring, performed outside of labor and followed by delivery at the same institution. Only cases resulting in live births were considered. Pregnancies were included if they were complicated by at least one of the following conditions: gestational diabetes mellitus, intrahepatic cholestasis of pregnancy, pregnancy-induced hypertension, or fetal growth restriction, classified as a primarily fetal disorder. Exclusion criteria encompassed multiple gestations, major congenital anomalies, incomplete clinical data, and poor-quality recordings with signal loss exceeding 20%. cCTG data were obtained using the OmniviewSisPorto Central—Fetal Monitoring Software, Version 4.0.15. The automatically computed parameters were exported and compiled into a dedicated database, which was subsequently enriched with perinatal information, including delivery outcomes and neonatal hospitalization duration.

The dataset incorporated a comprehensive set of cCTG-derived parameters, including signal loss, signal quality, baseline fetal heart rate (FHR), number of accelerations, fetal movements, uterine contractions, short-term variability (STV), percentage of STV values below 1 bpm, average STV, percentage of long-term variability (LTV) values below 5 bpm, and multiple deceleration-related indices, including total number of decelerations, repetitive decelerations, late decelerations, prolonged decelerations, and decelerations lasting more than five minutes. The automatic classification variable generated by the SisPorto software was excluded, as it was not relevant to the predictive modeling objectives of this study. In addition to cCTG-derived parameters, the database included gestational age at the time of monitoring and gestational age at delivery, both expressed in days, as well as binary clinical indicators reflecting the presence (1) or absence (0) of specific pregnancy-related pathologies. To simplify modeling and improve generalization, conditions with similar pathophysiological mechanisms were aggregated into unified categories—for instance, gestational and preexisting diabetes were grouped under “gestational diabetes (GD)”, and pregnancy-induced and preexisting hypertension were combined into “hypertension (HTA)”. The final set of binary variables, therefore included gestational diabetes (GD), hypertension (HTA), intrahepatic cholestasis of pregnancy (cholestasis), and intrauterine growth restriction (IUGR).

All recordings were anonymized prior to analysis. Signal quality was systematically evaluated, and only traces with signal loss not exceeding 20% were retained, while incomplete or corrupted recordings were excluded to ensure data integrity. The selection of analytical parameters was guided by both physiological rationale and previously validated evidence from the literature.

STV and LTV were included as principal indices of fetal autonomic regulation, reflecting the balance between sympathetic and parasympathetic modulation of cardiac activity and their sensitivity to changes in fetal oxygenation [12,17]. Baseline fetal heart rate, accelerations, and decelerations were incorporated as clinically interpretable manifestations of fetal well-being and adaptive response to intrauterine stimuli, whereas uterine activity and fetal movements were retained to account for transient physiological influences on heart rate dynamics [1,9,23,24,34,45,46]. This integrated set of parameters was deliberately chosen to capture both tonic and phasic components of fetal cardiovascular control, thereby providing a physiologically meaningful and statistically robust foundation for regression-based modeling of neonatal outcomes. The dataset contained no missing values; therefore, no imputation or preprocessing for missing data was necessary. This allowed all clinically relevant cCTG and perinatal features to be fully utilized in model development.

### 2.2. Dataset Preparation and Splitting

The dataset employed in this study comprised 694 high-risk singleton pregnancies and initially included a broad range of maternal, obstetric, and fetal variables derived from both electronic clinical records and cCTG monitoring. All records were fully anonymized prior to analysis to ensure data privacy and compliance with ethical research standards. After applying predefined quality control criteria, excluding incomplete or inconsistent cases, and performing data preprocessing, a subset of 20 clinically and technically relevant features was retained for model development. The feature selection process systematically removed redundant or weakly informative variables to ensure that the final predictors were physiologically meaningful and representative of key maternal and fetal conditions. Selection was guided by established associations with obstetric and neonatal outcomes, as well as by statistical relevance within the study cohort. The resulting 20 predictors were then used to train and evaluate four regression-based ML algorithms, with the duration of neonatal hospitalization (in days) defined as the target outcome.

The predictor variables included both categorical (e.g., presence of gestational diabetes, preexisting diabetes mellitus, intrahepatic cholestasis of pregnancy, pregnancy-induced hypertension, preexisting hypertension and fetal growth restriction) and continuous measures (e.g., gestational age at monitoring and delivery, fetal heart rate baseline, STV, LTV, number of accelerations and decelerations, duration of high- and low-variation episodes, percentage of time in high and low variation, uterine contraction frequency, total signal loss, and other signal quality metrics). Descriptive statistics for a subset of representative variables from the training dataset are presented in Table 1.

The binary clinical variables (HTA, IUGR, GD, and cholestasis) exhibited prevalence rates between 19% and 32%. Gestational ages at both monitoring and delivery were consistent with third-trimester pregnancies, while signal quality indices were uniformly high (median above 95%), confirming the technical reliability of the dataset. The target variable, defined as the number of days of neonatal hospitalization, displayed considerable variability, reflecting the clinical heterogeneity of the study population. Most newborns were discharged within seven days, corresponding to the expected duration of uncomplicated postnatal care, whereas a smaller subset required extended hospitalization due to perinatal complications. Consequently, the overall distribution of neonatal length of stay was discrete and right-skewed, with values extending up to 81 days (Figure 1).

This unbalanced distribution mirrors real-world clinical patterns, in which the majority of neonates experience short or moderate hospital stays, and only a limited number require prolonged medical care. To address potential bias arising from this imbalance, model training and validation were based on performance metrics robust to skewed outcomes—namely, the mean absolute error (MAE), coefficient of determination (*R*^2^), and root mean squared error (RMSE). Moreover, tree-based ensemble methods such as Random Forest, CatBoost, XGBoost, and LightGBM were selected because of their relative insensitivity to uneven target distributions and outliers, as they partition the feature space non-linearly and rely on median-based decision boundaries rather than means. Extreme outliers in hospitalization length were carefully reviewed and retained, since these cases represent clinically meaningful conditions (e.g., severe neonatal distress or growth restriction) and their removal could artificially simplify the prediction problem.

To ensure robust model development and unbiased performance estimation, the dataset was randomly divided into three subsets: a training set (508 records, ~73.2%) used for model fitting, a validation set (125 records, ~18.0%) employed for hyperparameter tuning and model selection, and a test set (61 records, ~8.8%) reserved exclusively for final performance evaluation. Given the continuous nature of the outcome variable, random splitting was performed without stratification. To enhance reproducibility across experiments, a fixed random seed was consistently applied.

As illustrated in Figure 2, the distributions of neonatal hospital stay (in days) across the training, validation, and test subsets exhibit highly similar right-skewed patterns. The majority of cases are concentrated within the first 10 days of hospitalization, with a small number of prolonged stays extending beyond 30 days. This similarity indicates that the random partitioning procedure successfully preserved the original outcome distribution across all subsets, ensuring that each subset remains representative of the overall data.

The external test set was curated to serve as a fully unseen evaluation benchmark. This dataset was withheld throughout the entire model development and validation process and was only accessed during the final assessment phase to evaluate the generalization capability of the best-performing model.

### 2.3. Machine Learning Methods

Four advanced machine learning regressors, Random Forest Regressor (RFR), CatBoost, XGBoost, and LightGBM, were applied to the dataset, and their performance was evaluated against a linear regression benchmark to contextualize model improvements over a standard approach. The Random Forest Regressor is an ensemble learning method that constructs multiple decision trees on bootstrapped subsets of data and averages their predictions to improve stability and reduce overfitting. CatBoost (Categorical Boosting), is a gradient boosting algorithm specifically optimized for handling categorical variables efficiently through ordered boosting and symmetric tree structures. XGBoost (eXtreme Gradient Boosting) is a highly efficient and scalable implementation of gradient boosting that leverages second-order gradient information and regularization to enhance predictive accuracy and control model complexity. LightGBM (Light Gradient Boosting Machine), is another gradient boosting framework designed for high efficiency on large-scale data, using histogram-based feature binning and leaf-wise tree growth to accelerate computation and reduce memory usage. Collectively, these algorithms represent robust and powerful approaches for non-linear regression tasks, offering a balance between predictive performance, computational efficiency, and generalization capability.

## 3. Results

This section presents the experimental outcomes and predictive performance obtained by training and evaluating the four ML regression models: Random Forest Regressor (RFR), Categorical Boosting (CatBoost), eXtreme Gradient Boosting (XGBoost) and Light Gradient Boosting Machine (LightGBM) compared against a benchmark linear regression model. The models were first trained, validated and fine-tuned, after which their performance was further tested to assess generalization capability.

### 3.1. Performance on Training and Validation Sets

The predictive capabilities of the ML regression models were systematically assessed on the training and validation subsets. Evaluation metrics included the coefficient of determination (*R*^2^, Equation (1)) as the primary indicator of model fit, supplemented by root mean squared error (RMSE, Equation (2)) and mean absolute error (MAE, Equation (3)) to ensure robustness and interpretability.(1)$$R^{2}=1-\frac{\sum_{i=1}^{n}{\left(y_{i}-\hat{y_{i}}\right)}^{2}}{\sum_{i=1}^{n}{\left(y_{i}-\bar{y}\right)}^{2}}$$(2)$$RMSE=\frac{1}{n}\sum_{i=1}^{n}{\left(y_{i}-\hat{y_{i}}\right)}^{2}$$(3)$$MAE=\frac{1}{n}\sum_{i=1}^{n}\left|y_{i}-\hat{y_{i}}\right|$$
where *n* is the number of data, $y_{i}$ is the actual (observed) value of output data (hospitalization days), $\hat{y_{i}}$ is the predicted value and $\bar{y}$ is the mean of output data. The coefficient of determination (*R*^2^) was used to assess the goodness of fit of the regression models. *R*^2^ quantifies the proportion of variance in the dependent variable that is explained by the model, with values closer to 1 indicating a better fit. In general, *R*^2^ values above 0.8 are considered indicative of a strong relationship between predicted and observed outcomes, whereas values between 0.5 and 0.8 reflect a moderate but acceptable level of predictive performance. In medical research, where biological variability and measurement uncertainty are often substantial, *R*^2^ values in the range of 0.5–0.8 can still represent clinically meaningful models, provided that predictions remain consistent and interpretable [47]. It is also important to note that a high *R*^2^ alone does not guarantee robust predictive performance; therefore, complementary metrics such as the Root Mean Squared Error (RMSE) and Mean Absolute Error (MAE) were also evaluated to provide a more comprehensive assessment of model accuracy and generalizability.

A systematic hyperparameter optimization procedure was employed to enhance model performance and ensure fair comparison across algorithms. The optimal model was selected as the one that maximized the mean coefficient of determination$$R_{mean}^{2}=\frac{R_{train}^{2}+R_{val}^{2}}{2},$$
reflecting the overall predictive capability across both training and validation datasets.

A classical Grid Search approach was used for the Random Forest Regressor, systematically exploring predefined parameter combinations to identify the optimal configuration. The search space was carefully designed to balance computational efficiency with thorough exploration. Specifically, the number of estimators (n_estimators) was varied from 30 to 100 in increments of 1 to capture a wide range of ensemble sizes. The minimum number of samples required at a leaf node (min_samples_leaf) ranged from 1 to 15 to explore different levels of tree granularity. The number of features considered when looking for the best split (max_features) was tested from 1 to 20, corresponding to the full range of available input features. To ensure reproducibility across experiments, the random_state parameter was fixed at 42.

The optimal configuration identified through Grid Search for the Random Forest model was: n_estimators = 92, min_samples_leaf = 1, max_features = 13, and random_state = 42, which achieved the best trade-off between bias and variance, ensuring stable and generalizable predictive performance on the validation data. All other hyperparameters were kept at their default values as implemented in the scikit-learn Python library.

Following the Random Forest optimization, the three gradient boosting algorithms (CatBoost, XGBoost, and LightGBM) were fine-tuned using the Optuna Python framework to maximize predictive performance on both training and validation datasets. For each model, 50 optimization trials were conducted, employing Bayesian optimization principles to efficiently explore the hyperparameter space and identify the best-performing configurations. This systematic procedure focused on achieving high predictive accuracy while minimizing overfitting, ultimately resulting in balanced model setups that optimized the trade-off between complexity and generalization capability.

CatBoost Regressor was fine-tuned using Optuna’s Bayesian optimization strategy, which explored 50 trials across a carefully defined hyperparameter space. The search included the number of iterations (100 to 1000), tree depth (3 to 10), learning rate (ranging logarithmically from 0.0001 to 0.1), and L2 leaf regularization (1.0 to 10.0). The random seed was fixed at 42 to ensure reproducibility, and verbosity was disabled during training. Each trial evaluated model performance using the mean coefficient of determination across training and validation sets, with a minimum threshold of Train *R*^2^ ≥ 0.9 and Val *R*^2^ ≥ 0.8 to ensure robust generalization. The optimal configuration identified by Optuna was: iterations = 279, depth = 7, learning_rate = 0.0134, and l2_leaf_reg = 5.72.

In the case of the XGBoost Regressor, a similar Optuna-based optimization strategy was applied, also spanning 50 trials. The hyperparameter search space included the number of estimators (100 to 1000), maximum tree depth (3 to 10), learning rate (0.0001 to 0.3, log-scaled), subsample ratio (0.5 to 1.0), column sampling ratio per tree (0.5 to 1.0), and regularization parameters for both L1 (reg_alpha) and L2 (reg_lambda) penalties, each ranging from 0.001 to 10.0. The random seed was again fixed at 42. Trials were evaluated using the same mean *R*^2^ criterion and minimum performance thresholds. The best-performing configuration was: n_estimators = 712, max_depth = 5, learning_rate = 0.00124, subsample = 0.93, colsample_bytree = 0.76, reg_lambda = 0.03, and reg_alpha = 0.92.

For the LightGBM Regressor, the optimization process followed the same Bayesian approach using Optuna with 50 trials to explore a rich hyperparameter space. Parameters included the number of estimators (100 to 1000), maximum depth (3 to 10), learning rate (0.0001 to 0.3, log-scaled), number of leaves (20 to 300), feature fraction (0.5 to 1.0), bagging fraction (0.5 to 1.0), and both L1 and L2 regularization terms (reg_alpha and reg_lambda, each from 0.001 to 10.0). As with the other models, the random seed was set to 42 for consistency. The optimization objective was to maximize the mean *R*^2^ across training and validation sets, subject to the same minimum performance constraints. The final configuration was: n_estimators = 666, max_depth = 10, learning_rate = 0.090901, num_leaves = 84, feature_fraction = 0.944701, bagging_fraction = 0.917056, reg_lambda = 0.590050, and reg_alpha = 0.001139.

The optimal configuration identified for each model was used to train the final version, resulting in consistently strong predictive performance across training, validation, and test datasets.

For benchmarking, a linear regression model with Ridge regularization was implemented. Ridge applies L2 regularization to penalize large coefficients and improve stability, providing an interpretable baseline against which the performance of advanced ML regressors can be evaluated.

A summary of the results is presented in Table 2, where the metrics are presented for the best regression models.

Among the tested algorithms, the Random Forest Regressor exhibited the most balanced and consistent performance, achieving an *R*^2^ of 0.94 on the training set and 0.80 on the validation set. This alignment between training and validation scores suggests that the model effectively captured complex non-linear relationships while maintaining generalization capacity, with a mean *R*^2^ exceeding 0.87. In contrast, both CatBoost and LightGBM showed evidence of overfitting. CatBoost achieved a perfect fit on the training data (*R*^2^ = 1.00), yet its validation performance dropped to 0.73. Similarly, LightGBM yielded an inflated training score (*R*^2^ = 0.99) but a comparatively lower validation score (*R*^2^ = 0.67). These discrepancies imply that both models may have over-learned patterns specific to the training data, thereby limiting their ability to generalize to unseen cases. XGBoost demonstrated an intermediate profile, with an *R*^2^ of 0.99 on the training set and 0.81 on the validation set. While it outperformed CatBoost and LightGBM on validation, the gap between training and validation scores still indicated a moderate degree of overfitting. Nevertheless, its average performance (mean *R*^2^ = 0.9) ranked second overall, following Random Forest.

In contrast, the linear regression benchmark with Ridge regularization exhibited substantially lower predictive power, achieving a training *R*^2^ of 0.5910, validation *R*^2^ of 0.5489, and a mean *R*^2^ of 0.5699. These results demonstrate that all four ML models outperform the conventional linear approach, highlighting their ability to capture complex, non-linear relationships present in the cCTG features. The superior performance of ensemble-based ML regressors suggests that non-linear interactions among cCTG parameters are important for accurately predicting neonatal length of stay.

Based on generalization performance and train–validation consistency, the ML models can be ranked as follows (Figure 3):

**Random Forest**: Highest generalization ability and most balanced performance.**XGBoost**: Strong predictive accuracy with moderate overfitting.**CatBoost**: Perfect training fit but reduced validation performance.**LightGBM**: Low validation score and largest performance gap.**Linear regression**: Poor performance both on train and validation.

These findings underscore the robustness of the Random Forest Regressor in predicting neonatal hospitalization duration, offering an optimal trade-off between bias and variance when compared to gradient boosting methods. To further assess the predictive accuracy of the Random Forest model, additional performance metrics were computed, namely the Root Mean Squared Error (RMSE) and the Mean Absolute Error (MAE). These metrics provide complementary insights: RMSE emphasizes larger errors due to its quadratic penalization, whereas MAE reflects the average magnitude of prediction errors in a more interpretable manner. A detailed summary of the Random Forest performance across training and validation is presented in Table 3.

The results confirm that Random Forest maintained high explanatory power (*R*^2^ = 0.94 on training and 0.80 on validation) while keeping prediction errors at clinically acceptable levels. The relatively low MAE across all datasets suggests that, on average, the predicted hospitalization length differed by fewer than two days from the observed values, which is meaningful in a clinical context. The moderate increase in RMSE and MAE on the validation and test sets was expected, reflecting real-world variability and case complexity. Importantly, the consistent generalization performance indicates that the Random Forest model provides a reliable framework for predicting neonatal hospitalization duration.

All experiments were conducted on a workstation equipped with an Intel^®^ Core™ i3-1005G1 CPU (1.20 GHz, dual-core), 12 GB RAM, running Windows 10 (64-bit). Model implementation and evaluation were performed in Python 3.11.9.

The following software packages and versions were used to ensure reproducibility: scikit-learn 1.4.1.post1 (for Random Forest Regressor, Linear regression with Ridge regularization, GridSearchCV with 5-fold cross-validation, train_test_split, Standard-Scaler, and evaluation metrics including *R*^2^-score, mean_squared_error, and mean_absolute_error), CatBoost 1.2.8 (CatBoostRegressor), XGBoost 3.0.2 (XGBRegressor), LightGBM 4.6.0 (LGBMRegressor), and Optuna 4.4.0 (for hyperparameter optimization of gradient boosting models with 50 trials per model). Data preprocessing and manipulation utilized pandas 2.2.1 for structured data operations and Excel file handling, and NumPy 1.26.4 for numerical computations. Model persistence was achieved using Python’s pickle module for serialization. Hyperparameter optimization strategies included exhaustive grid search via GridSearchCV for the linear regression model and Optuna for gradient boosting algorithms.

### 3.2. Performance on Test Set

The independent test set, comprising 61 patient records, was employed to evaluate the final generalization capabilities of all optimized models on previously unseen data. The evaluation was conducted using the same metrics employed during training and validation phases, with particular emphasis on the *R*^2^ coefficient of determination as the primary performance indicator.

The performance evaluation on the independent test set was conducted using the optimized configurations of each model, obtained following the hyperparameter tuning procedure performed during the training and validation phases. This approach ensures that the reported results accurately reflect the true generalization capacity of the final models. Only the best-performing, fine-tuned versions, those maximizing the mean *R*^2^ score across the training and validation subsets, were utilized for testing, thereby providing an objective and reliable assessment of predictive performance on unseen data.

The Random Forest Regressor demonstrated superior performance on the test set, achieving an *R*^2^ score of 0.8226, which represents the highest predictive accuracy among all evaluated models. This result confirms the model’s excellent generalization ability, with a slight performance improvement observed compared to the validation set (*R*^2^ = 0.80). This improvement from validation to test set is particularly encouraging, as it suggests the model generalizes very well to new, unseen data and indicates robust model stability.

The other ensemble methods showed varying degrees of performance on the independent test set. CatBoost achieved an *R*^2^ score of 0.7059, representing a substantial decrease from its validation performance (*R*^2^ = 0.7325). XGBoost obtained an *R*^2^ of 0.6911, showing the most pronounced performance degradation from validation (*R*^2^ = 0.8147) to the test set. LightGBM reached an *R*^2^ of 0.6851, maintaining relatively stable performance compared to its validation results (*R*^2^ = 0.6751). These results further consolidate the Random Forest Regressor as the most suitable approach for predicting newborn hospital stay duration in this clinical context.

The performance metrics indicate that the Random Forest model maintains consistent predictive power across different data distributions, suggesting its potential applicability in real-world clinical settings. The observed performance improvement from validation to test set (0.80 to 0.82) is particularly noteworthy, as it indicates that the test set characteristics align well with the model’s learned patterns and demonstrates notable generalization capability.

The scatter plot from Figure 4 shows a tight clustering of points around the diagonal, indicating strong agreement between predicted and actual hospitalization durations. Most samples fall close to the ideal prediction line (y = x), confirming high accuracy in the 0–10-day range, which represents the majority of cases. Deviations from the diagonal correspond to atypical clinical scenarios with extended stays. Overall, the Random Forest model demonstrates precise and stable predictive behavior across the observed range.

Figure 5 presents the histogram of the residual errors $y_{i}-\hat{y_{i}}$, calculated for the hospitalization days from the test set. From the figure one can observe that the residuals display a nearly symmetric, zero-centered distribution, indicating the absence of systematic bias in model predictions. Most residuals lie within ±3 days, reflecting small prediction errors and consistent performance. The mild right tail denotes a slight underestimation tendency for extreme outliers, newborns with very long hospitalizations, expected due to their low prevalence.

Figure 6 illustrates, on the left side, the histogram representing the distribution of hospitalization days in the test set, while the right panel displays the corresponding boxplot statistics for the same data. The test set distribution is clearly right-skewed, showing that most newborns were hospitalized for fewer than seven days. A small fraction of cases with stays exceeding 30 days defines a long right tail, associated with complex or severe clinical conditions. This asymmetry underscores the heterogeneous nature of the dataset and the inherent difficulty of predicting length of stay across diverse clinical profiles.

To further quantify the statistical reliability of the model’s performance on unseen data, two complementary resampling-based procedures were employed: bootstrap confidence estimation and a permutation-based significance test. The bootstrap procedure (*n* = 2000 resamples) involves repeatedly drawing random samples with replacement from the test set and recalculating the *R*^2^ for each sample. This process generates an empirical sampling distribution of the performance metric, from which a 95% confidence interval (CI) can be derived. Figure 7 presents the distribution of *R*^2^ values obtained through bootstrap resampling. The 95% confidence interval (CI) is indicated by the vertical dashed lines, while the red solid line marks the *R*^2^ computed on the test set. The obtained interval of [0.6744, 0.9089] reflects the uncertainty associated with the finite test sample and provides a more realistic estimate of how the model might perform on new, unseen data. Importantly, the relatively narrow interval centered around the observed *R*^2^ = 0.8226 supports the stability and reliability of the Random Forest predictions.

In contrast, the permutation test addresses a different question, whether the observed predictive performance could have occurred merely by chance. By randomly shuffling the outcome variable (length of stay) across test samples and re-evaluating the model multiple times (*n* = 2000 permutations), a null distribution of *R*^2^ values is generated, corresponding to the scenario where no real relationship exists between predictors and target. Figure 8 illustrates the distribution of *R*^2^ values obtained from the permutation test, with the red vertical line indicating the observed *R*^2^ from the test set. As shown, the test *R*^2^ exceeds all permuted values, corresponding to a *p*-value of <0.001, thereby confirming that the model’s predictive performance is highly unlikely to have occurred by chance. The resulting *p*-value < 0.001 indicates that none of the randomized permutations achieved an *R*^2^ comparable to the observed one, confirming that the model’s predictive accuracy is highly significant and not an artifact of random data alignment.

Together, these two complementary validation strategies—bootstrap and permutation testing—provide a rigorous statistical framework for evaluating model generalization on limited clinical datasets. While the bootstrap quantifies performance uncertainty, the permutation test establishes statistical significance relative to random expectation, ensuring that the reported test results are both robust and meaningful for clinical interpretation.

### 3.3. Summary of Findings

The comprehensive experimental evaluation demonstrated that the Random Forest Regressor consistently outperformed all other regression techniques across training, validation, and independent test sets. It achieved excellent performance metrics, with *R*^2^ scores of 0.94 for the training set, 0.80 for the validation set, and 0.82 for the test set, yielding an average predictive performance of 0.87 across the modeling phases.

These results underscore the robustness and strong generalization capacity of the Random Forest model for this clinical dataset. The consistent performance across multiple data splits confirms its reliability in predicting neonatal length of stay. Notably, the slight improvement observed from validation (0.80) to test (0.82) suggests that the model performs well when applied to unseen patient data, indicating stable external validity.

From a clinical perspective, these findings demonstrate that cCTG-derived parameters, when integrated with maternal and perinatal variables, can yield meaningful predictive insights into postnatal adaptation and neonatal morbidity risk. The accurate estimation of hospital stay duration serves as a surrogate marker of overall neonatal health, as prolonged admissions often indicate respiratory distress, feeding difficulties, or metabolic instability. Early antenatal identification of high-risk cases could therefore facilitate timely clinical intervention, optimize neonatal intensive care resource allocation, and enhance communication with families regarding expected postnatal outcomes.

Importantly, the results further support that fetal heart rate variability patterns captured through cCTG are not only markers of intrapartum well-being but also indicators of antenatal physiological adaptation and resilience, with clear implications for neonatal outcomes. Variability indices such as STV and LTV appear to carry prognostic value regarding the fetus’s capacity to tolerate hypoxic or metabolic stress. The strong predictive accuracy of the model reinforces the concept that computerized analysis of fetal cardiac dynamics can objectively quantify aspects of autonomic regulation that are clinically significant yet often imperceptible through traditional visual interpretation.

By assisting clinicians in determining the optimal timing of delivery—particularly in pregnancies complicated by hypertension, diabetes mellitus, intrahepatic cholestasis, or fetal growth restriction—such predictive models may contribute to balancing the risks of prematurity against those of intrauterine compromise. Early prediction of potential postnatal complications could also inform decisions on corticosteroid administration, in utero transfer to tertiary care centers, and intensified antenatal surveillance, all of which are pivotal for improving perinatal outcomes.

Integrating data-driven decision support systems based on cCTG analysis into obstetric workflows could therefore enhance both the safety and efficiency of perinatal care. Anticipating neonatal care needs before birth enables better resource planning, reduces emergency interventions, and fosters closer collaboration between obstetric, neonatal, and anesthetic teams. Moreover, optimizing delivery timing in high-risk pregnancies could help shorten neonatal hospitalization, reduce healthcare costs, and ultimately improve maternal–fetal outcomes.

While other ensemble methods also demonstrated competitive training performance—particularly CatBoost, which achieved a perfect training *R*^2^ of 1.00—their pronounced decline on test data suggested overfitting. In contrast, the Random Forest model effectively mitigated this risk through its bagging mechanism and random feature selection, ensuring a superior balance between bias and variance.

Figure 9 summarizes the Random Forest model’s performance across the training, validation, and independent test datasets using three complementary evaluation metrics. The coefficient of determination (*R*^2^) quantifies the proportion of variance in neonatal length of stay explained by the model and remained consistently high across all splits (Training: 0.9428; Validation: 0.8042; Test: 0.8226), indicating a strong overall fit and stable generalization capability. The Root Mean Square Error (RMSE)—which emphasizes larger deviations—was 2.146, 3.218, and 3.409 days for the training, validation, and test sets, respectively, reflecting a moderate and expected increase on unseen data. The Mean Absolute Error (MAE), representing the average prediction deviation in days, remained low (Training: 0.901; Validation: 1.890; Test: 2.018), confirming clinically acceptable accuracy for neonatal care planning. Collectively, these metrics demonstrate that the Random Forest model achieves a favorable balance between predictive precision and generalization robustness from development to external evaluation.

Table 4 presents the comparative performance of the four ensemble regression models evaluated in this study. The Random Forest Regressor achieved the highest predictive accuracy (*R*^2^ = 0.8226), followed by CatBoost (0.7059), XGBoost (0.6911), and LightGBM (0.6851). Random Forest was therefore selected as the final predictive model, given its superior generalization performance and methodological stability. It exhibited the smallest discrepancy between training, validation, and test results (*R*^2^ = 0.9428 → 0.8042 → 0.8226) and was the only algorithm to demonstrate a performance improvement from the validation to the test phase. This consistent pattern indicates robust generalization to unseen data, whereas gradient boosting models achieved near-perfect fits on training sets (*R*^2^ ≈ 1.0) but displayed notable performance declines during external validation, suggesting a higher risk of overfitting.

From a methodological standpoint, Random Forest offers a particularly favorable bias–variance trade-off. Its intrinsic bootstrap aggregation and random feature sub-sampling mechanisms effectively mitigate variance without the need for extensive regularization. These characteristics are especially advantageous when working with a moderately sized dataset (*n* = 694) containing mixed binary and continuous predictors, noise-prone cCTG-derived variables, and potential inter-feature correlations.

Another key factor supporting the selection of Random Forest is its robustness to preprocessing assumptions. Unlike boosting-based approaches, it can naturally accommodate heterogeneous feature scales and non-linear relationships, eliminating the need for prior data normalization or monotonic constraints. This property aligns with the study’s inference pipeline, which operates directly on raw clinical data, thereby reducing the risk of distributional drift between training and test phases.

Finally, the optimized Random Forest configuration (n_estimators = 92; max_features = 13; min_samples_leaf = 1) achieved high predictive performance with minimal hyperparameter tuning and exhibited low sensitivity to parameter variations. In contrast, the boosting algorithms required extensive regularization yet continued to show larger generalization gaps. Taken together, these findings confirm Random Forest as the most reliable, interpretable, and practically applicable model for predicting neonatal hospitalization duration based on cCTG-derived and clinical parameters.

The optimized Random Forest model, selected for its superior generalization capability, has been integrated into a clinician-oriented application that provides real-time predictions of neonatal length of stay. The workflow begins with case initialization, followed by structured data entry organized into clinically relevant categories: pregnancy-related characteristics (gestational age at monitoring and delivery), binary indicators of maternal or fetal pathology—gestational diabetes (GD), hypertension (HTA), intrahepatic cholestasis of pregnancy (cholestasis), and intrauterine growth restriction (IUGR)—and cCTG-derived parameters including signal loss, signal quality, baseline fetal heart rate (FHR), accelerations, fetal movements, uterine contractions, short-term variability (STV) indices (percentage of STV values < 1 bpm and average STV), long-term variability (LTV) indices (percentage of LTV values < 5 bpm), and deceleration-related metrics (total, repetitive, late, prolonged, and those exceeding five minutes).

To ensure data integrity and consistency, the interface incorporates mandatory input fields, automated validation, and plausibility checks that flag incomplete or physiologically implausible entries. Upon submission, a one-click execution applies the pre-trained Random Forest model and instantly generates a prediction of neonatal hospitalization duration, accompanied by a confidence indicator derived from test-set performance metrics. The output is displayed in an intuitive, interpretable format with visual cues designed to facilitate rapid clinical interpretation.

A patient-specific report can be automatically generated for inclusion in the electronic medical record or multidisciplinary team discussions. In parallel, anonymized data and predictions can be securely stored or exported for auditing, quality assurance, and research purposes. Developed to support rather than replace clinical judgment, the system prioritizes transparency, traceability, and responsible use, offering clinicians a robust, data-driven decision support tool to enhance resource planning, neonatal bed management, and patient–family counseling in high-risk pregnancies.

## 4. Discussion

The accurate prediction of neonatal outcomes remains one of the most challenging goals in perinatal medicine, particularly in pregnancies complicated by maternal or fetal disorders. Computerized cardiotocography (cCTG) offers an objective, quantifiable evaluation of fetal heart rate and uterine activity, providing insight into autonomic regulation and oxygenation dynamics. In the present study, regression-based machine learning (ML) techniques were applied to cCTG recordings obtained outside of labor to estimate the duration of neonatal hospitalization—an objective, continuous, and clinically meaningful measure of postnatal adaptation. This approach bridges the gap between fetal surveillance and neonatal outcome prediction, introducing a data-driven framework for perinatal risk assessment.

Among the tested algorithms, the Random Forest (RF) model achieved the highest predictive performance, with a test *R*^2^ of 0.8226, RMSE of 3.41 days, and MAE of 2.02 days, outperforming CatBoost (*R*^2^ = 0.7059), XGBoost (*R*^2^ = 0.6911) and LightGBM (*R*^2^ = 0.7871). The RF model also demonstrated excellent generalization across the training, validation, and test sets (0.9428 → 0.8042 → 0.8226), confirming its robustness and low susceptibility to overfitting. Linear regression, by contrast, showed limited predictive ability (*R*^2^ = 0.5699), highlighting the advantage of ensemble-based ML models for complex, non-linear relationships in perinatal data.

These findings are broadly consistent with recent neonatal outcome prediction studies. Regression-based ML approaches applied to large NICU datasets have reported *R*^2^ values ranging from 0.68 to 0.82 [36], while DL methods using time-series EHR data achieved similar performance [37].

The model’s mean prediction error—approximately two days—is both clinically interpretable and operationally acceptable, offering a reliable estimate of neonatal resource needs. In practice, short variations in one to two days in hospitalization length rarely influence clinical decision-making, as such differences commonly reflect non-critical factors such as monitoring of feeding and weight gain, brief phototherapy for mild hyperbilirubinemia, or routine observation prior to discharge. Consequently, this level of predictive precision mirrors real-world variability and demonstrates the model’s capacity to estimate a continuous, multifactorial outcome with high fidelity. Such performance is sufficient to support anticipatory planning, optimize neonatal resource allocation, and facilitate individualized perinatal management, without increasing the risk of clinical misclassification or inappropriate intervention.

Overall, the findings of this study highlight the strong predictive capability of regression-based ensemble models in quantifying neonatal outcomes from antepartum cCTG data. The Random Forest algorithm effectively captured intricate, non-linear interactions between fetal heart rate variability metrics, uterine activity patterns, and maternal indicators such as gestational age and comorbidities. This integrative capacity underscores the model’s ability to detect subtle physiological signatures predictive of neonatal adaptation after birth. By converting high-dimensional physiological data into a continuous numerical outcome, the proposed framework enhances the granularity of perinatal assessment and provides an interpretable, data-driven tool for objective neonatal prognosis.

To better understand the factors driving the model’s predictions, a feature importance analysis was performed using the Random Forest algorithm, thereby enhancing the interpretability and clinical relevance of the proposed framework. Gestational age at delivery emerged as the dominant predictor, explaining 56.9% of the model’s decision variance—a result fully consistent with established clinical knowledge, as prematurity is a well-recognized determinant of prolonged neonatal hospitalization. Although this variable exerted the strongest influence, the model also identified several additional maternal, fetal, and perinatal contributors. Gestational age at the time of monitoring ranked second, accounting for 13.2% of the total variance, while multiple cCTG-derived parameters—such as fetal movements (4.1%), the percentage of short-term variability (STV) values below 1 bpm (1.9%), average STV (1.3%), and the percentage of long-term variability (LTV) values below 5 bpm (1.3%)—also demonstrated meaningful contributions. These findings align with established physiological evidence that reduced STV and LTV, along with pathological decelerations, reflect impaired autonomic regulation and diminished fetal oxygenation capacity.

Collectively, these results demonstrate that the predicted duration of neonatal hospitalization is influenced not only by gestational maturity but also by fetal autonomic function as reflected in cCTG patterns recorded outside of labor. This emphasizes the dual contribution of developmental stage and physiological resilience to postnatal adaptation. Anticipating the duration of neonatal hospitalization thus represents a valuable objective in both obstetric and neonatal care, serving as an indirect indicator of the newborn’s capacity for extrauterine adjustment. Predictive information derived from antenatal cCTG recordings can assist clinicians in optimizing the timing of delivery, allocating neonatal intensive care resources, and counseling families regarding expected postnatal trajectories. Importantly, since all recordings were obtained outside the intrapartum period, they capture baseline fetal physiology without the transient hypoxic fluctuations induced by uterine contractions, thereby offering a more stable and unbiased reflection of fetal well-being.

Compared with conventional fetal surveillance methods that rely on qualitative or categorical CTG interpretation, this approach introduces a quantitative, regression-based framework for outcome prediction. By estimating a continuous variable—neonatal length of stay—rather than discrete risk categories, the model represents a paradigm shift from reactive to predictive obstetrics. Through this transition, data-driven algorithms can provide anticipatory insights into neonatal prognosis and resource needs. Moreover, by focusing on high-risk pregnancies complicated by gestational diabetes, intrahepatic cholestasis, hypertension, or fetal growth restriction, the model demonstrates the feasibility of applying AI-based decision support tools to clinically heterogeneous and complex patient populations.

This study has certain limitations that should be acknowledged. The investigation was performed within a single tertiary center, which may constrain the generalizability of the findings. Multicenter validation was not feasible, as most maternity hospitals in the country still employ conventional CTG systems, and few private institutions utilize the Omniview–SisPorto Central Fetal Monitoring Software (Version 4.0.15) used in this analysis. These differences in acquisition platforms and parameter definitions currently limit large-scale data harmonization. Nonetheless, the present dataset encompasses a broad and diverse spectrum of clinical conditions, gestational ages, and monitoring contexts. Recordings were obtained by multiple clinicians, capturing varied high-risk pregnancy profiles, which enhances the internal representativeness and supports the robustness of the model within the single-center framework. Institutional factors—such as discharge policies and local healthcare logistics—may still influence neonatal hospitalization independently of medical factors and should be considered when interpreting the results.

In addition to these methodological and institutional considerations, the dataset also exhibited distributional limitations. The outcome variable—neonatal length of stay—was right-skewed, with prolonged hospitalizations being relatively underrepresented. This reflects the natural distribution of clinical cases rather than sampling bias. While ensemble algorithms such as Random Forest, CatBoost, XGBoost, and LightGBM are inherently robust to unbalanced outcomes and outliers, a minor prediction bias toward the most frequent hospitalization range cannot be fully excluded. Extreme values were deliberately retained in the analysis, as they represent clinically meaningful scenarios (e.g., severe distress, prematurity, or growth restriction) rather than measurement artifacts. Future research, including larger, more balanced cohorts—or employing synthetic oversampling techniques for regression tasks—may further enhance generalization to rare and severe neonatal outcomes.

Finally, although cCTG parameters effectively capture essential aspects of fetal physiology, integrating additional maternal and fetal predictors could further refine model performance. Combining biochemical markers (e.g., serum bile acids, blood pressure dynamics, or glycemic control indices) with Doppler flow parameters and early neonatal metrics may enhance both predictive accuracy and physiological interpretability. Future efforts should thus prioritize multicenter validation, the incorporation of explainable ML architectures, and the integration of multimodal data sources to strengthen predictive precision and foster clinical translation. Ultimately, embedding such models within clinician-oriented decision support systems could meaningfully advance perinatal management by enabling earlier, individualized, and evidence-based interventions.

## Figures and Tables

**Figure 1 diagnostics-15-02964-f001:**
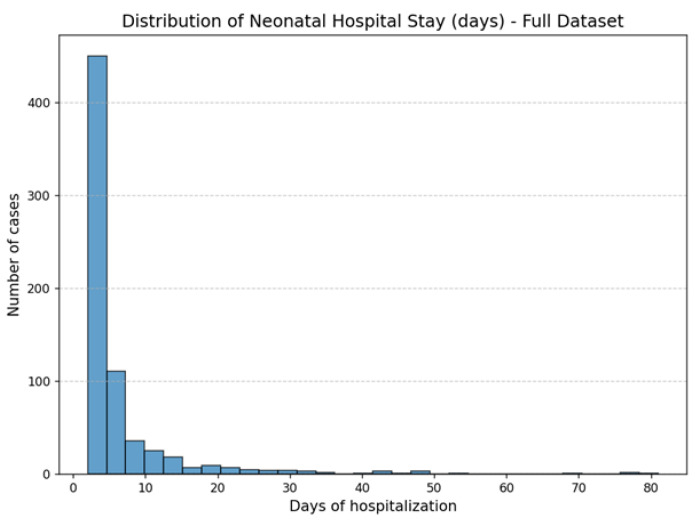
Distribution of the neonatal hospital stay (target value) in the entire dataset.

**Figure 2 diagnostics-15-02964-f002:**
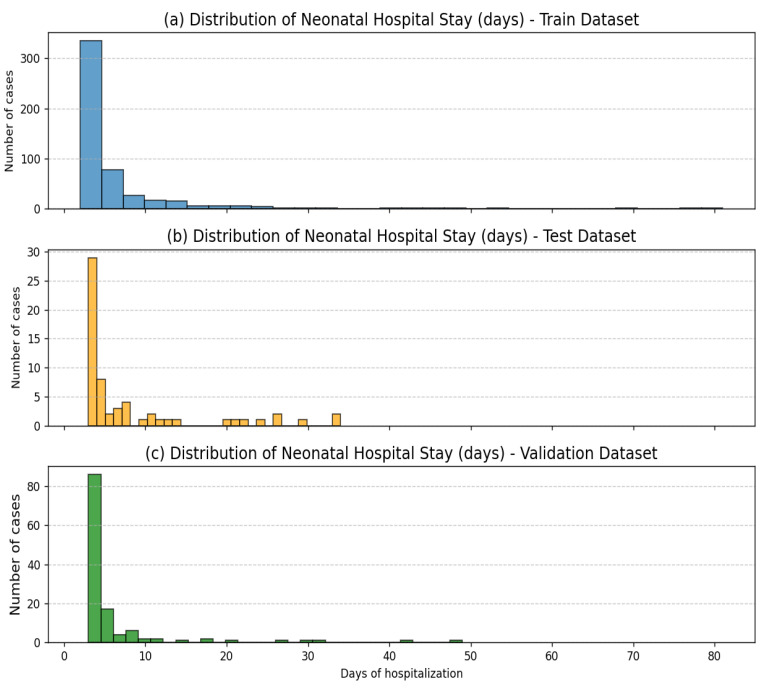
Distribution of neonatal hospital stay (days) across the (**a**) training, (**b**) validation, and (**c**) test subsets. All subsets display similar right-skewed distributions, confirming that the random splitting preserved the overall outcome structure of the dataset.

**Figure 3 diagnostics-15-02964-f003:**
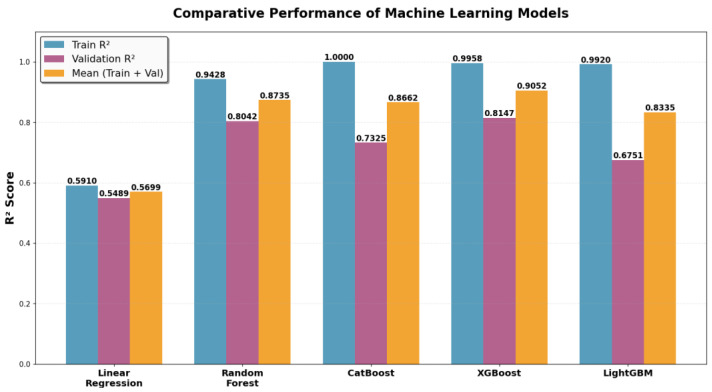
Comparative performance of ML models for neonatal length of stay prediction.

**Figure 4 diagnostics-15-02964-f004:**
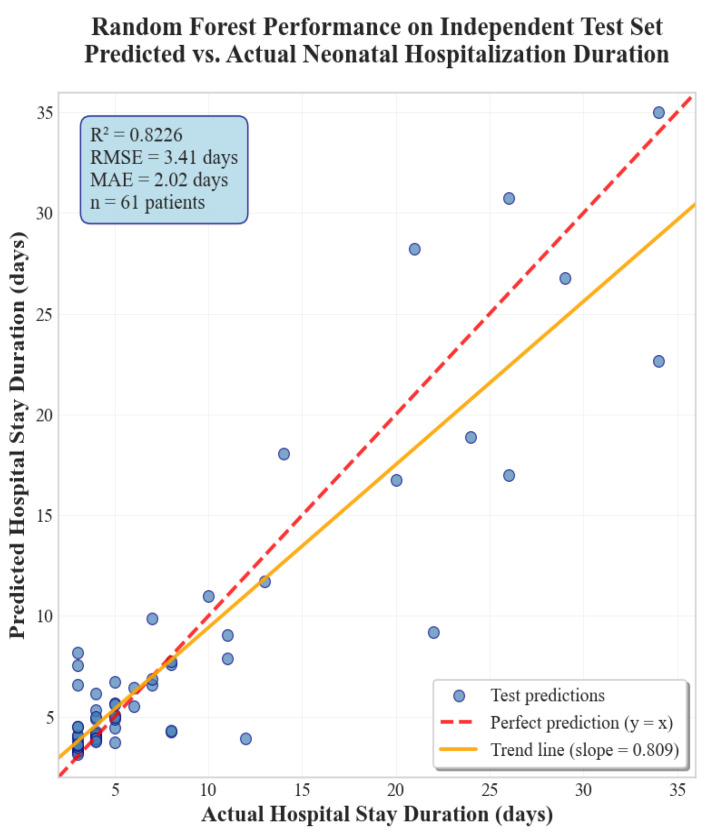
Scatter plot of predicted vs. actual values on the test set for Random Forest.

**Figure 5 diagnostics-15-02964-f005:**
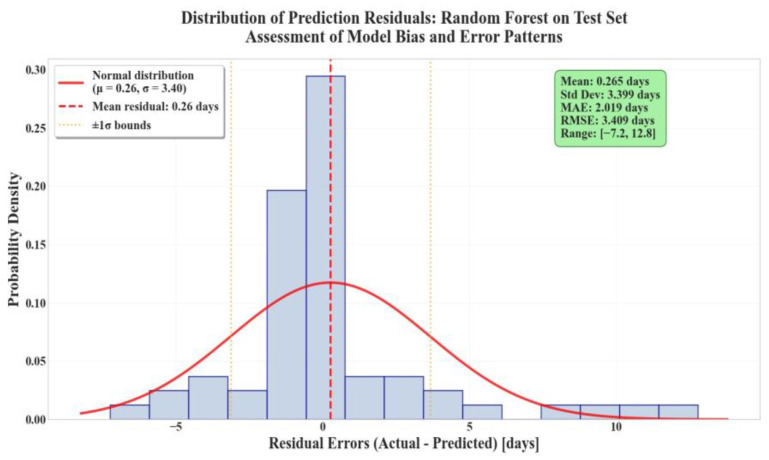
Histogram of residual errors for Random Forest on the test set.

**Figure 6 diagnostics-15-02964-f006:**
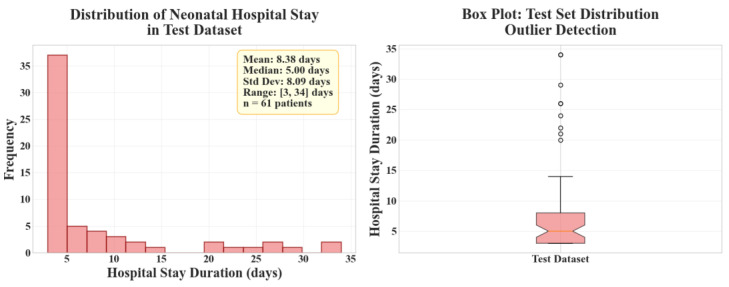
Test set distribution.

**Figure 7 diagnostics-15-02964-f007:**
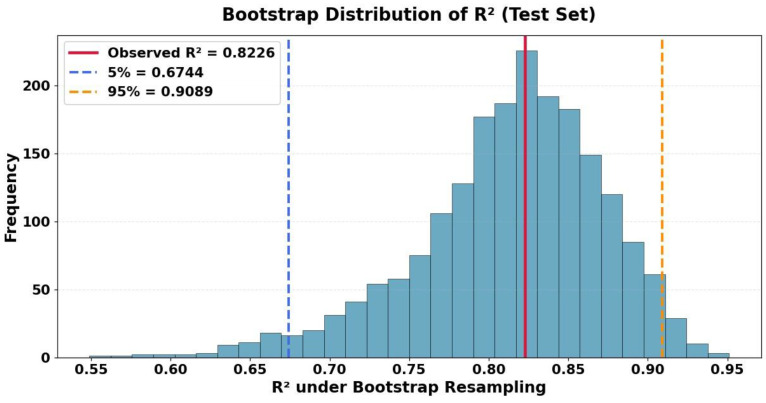
Histogram of *R*^2^ under Bootstrap resampling.

**Figure 8 diagnostics-15-02964-f008:**
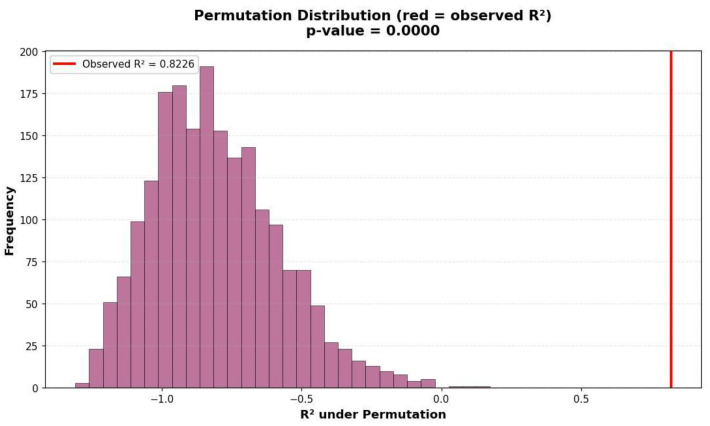
Histogram of *R*^2^ under permutation.

**Figure 9 diagnostics-15-02964-f009:**
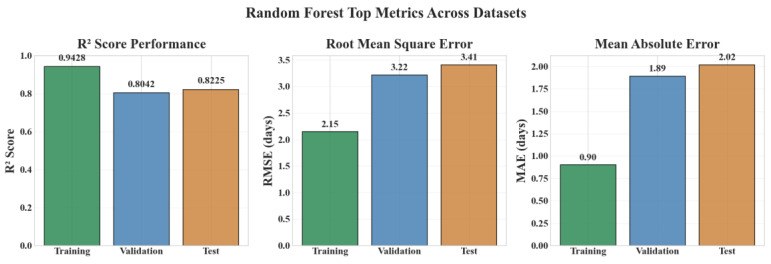
Random Forest top metrics across datasets (*R*^2^, RMSE, MAE).

**Table 1 diagnostics-15-02964-t001:** Descriptive statistics of selected variables in the training dataset.

Variable	Mean	Std	Min	Median	Max
HTA	0.32	0.46	0	0	1
IUGR	0.32	0.46	0	0	1
Gestational diabetes (GD)	0.27	0.44	0	0	1
Cholestasis	0.19	0.39	0	0	1
Gestational age at monitoring (days)	255.8	16.0	197	260	281
Gestational age at delivery (days)	263.0	12.9	200	266	285
Neonatal hospital stays (days, target variable)	6.57	8.65	2	4	81
Signal loss (%)	5.13	5.61	0	3	29
Signal quality (%)	94.99	5.47	71	97	100

**Table 2 diagnostics-15-02964-t002:** Performance of ML models on training and validation sets, reported for *R*^2^ scores.

Model	Train *R*^2^	Validation *R*^2^	Mean (Train + Val)
Random Forest	0.9428	0.8042	0.8735
CatBoost	1.0000	0.7325	0.8662
XGBoost	0.9958	0.8147	0.9052
LightGBM	0.9920	0.6751	0.8335
Linear Regression	0.5910	0.5489	0.5699

**Table 3 diagnostics-15-02964-t003:** Extended performance metrics for the Random Forest Regressor.

Dataset	*R* ^2^	RMSE	MSE
Training	0.9428	2.1456	0.9009
Validation	0.8042	3.2177	1.8904

**Table 4 diagnostics-15-02964-t004:** Performance of the top-performing models on the test set.

Model	Train *R*^2^	Validation *R*^2^	Test *R*^2^	Mean (Train + Val)
Random Forest	0.9428	0.8042	0.8226	0.8735
CatBoost	1.0000	0.7325	0.7059	0.8662
XGBoost	0.9958	0.8147	0.6911	0.9052
LightGBM	0.9920	0.6751	0.6851	0.8335

## Data Availability

The data underlying this study are not publicly available due to patient confidentiality and institutional data protection regulations. Aggregated or anonymized data may be made available from the corresponding author upon reasonable request and with permission of the Filantropia Clinical Hospital.

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
