# Peer review of "Diagnostics2025, 15(23), 2964;https://doi.org/10.3390/diagnostics15232964"

_diagnostics, 2025, doi:10.3390/diagnostics15232964_

Round 1

Reviewer 1 Report

Comments and Suggestions for Authors

The paper aims to predict the neonatal length of hospital stay using computerized cardiotocography data and maternal clinical variables in high-risk pregnancies. The authors trained four regression-based machine learning models, Random Forest, CatBoost, XGBoost, and LightGBM, on 694 cases. The Random Forest model achieved the best performance, suggesting its potential clinical utility for neonatal resource planning. The work proposes a non-invasive and data-driven approach to improve perinatal care.

The paper is clinically important, and the integration of ML with cCTG is innovative. The manuscript is generally well organized and methodologically clear. However, before acceptance, the paper requires major revision to improve scientific rigor, clinical interpretation, and clarity.

  1. External validation is missing. Results from a single-center dataset limit generalizability. The authors should validate the model on data from another institution or clearly discuss this limitation.

  2. Feature importance and explainability are not sufficiently analyzed. A clear explanation of which cCTG parameters most influence the prediction would improve clinical trust.

  3. Comparative baseline models (for example, linear regression or clinical scoring) should be included for benchmarking.

  4. Dataset imbalance and handling of extreme outliers need more discussion.

  5. The clinical meaning of a 2-day prediction error should be better interpreted. Does this difference significantly affect decision-making?

  6. Figures and tables are numerous; some could be simplified to enhance readability.

  7. The Discussion section should compare results with more recent neonatal outcome prediction studies and emphasize practical implementation limits.

Author Response

We thank Reviewer 1 for the careful and constructive evaluation of our manuscript. The comments provided were highly valuable in improving the methodological clarity, scientific rigor, and clinical relevance of our work. We have addressed all points in detail, and the revised version reflects these important contributions. The modifications carried out in the revised manuscript are in red.

Comments 1: External validation is missing. Results from a single-center dataset limit generalizability. The authors should validate the model on data from another institution or clearly discuss this limitation.

Response 1: We are grateful to the reviewer for this thoughtful remark, which helped us clarify this limitation. We fully acknowledge that the use of a single-center dataset represents a constraint regarding external generalizability. Multicenter validation could not be conducted at this stage, as most maternity hospitals in our country still employ conventional CTG systems. Currently, only Filantropia Hospital utilizes computerized CTG (cCTG) analysis on a regular basis. This work is part of the PhD project of Dr. Bianca Danciu, supervised by Dr. Anca Simionescu at the “Carol Davila” University of Medicine and Pharmacy, in collaboration with engineers and researchers in artificial intelligence.

The primary aim of this stage was not to perform external validation, but rather to conduct an in-depth analysis of real clinical data and explore their potential predictive value for neonatal outcomes using a deep-learning approach. The results presented here represent an intermediary phase of a broader, ongoing research project that will include subsequent steps of external validation and model refinement.

Moreover, very few institutions currently use the same Omniview-SisPorto Central Fetal Monitoring Software (Version 4.0.15) as in the present study, which makes it methodologically challenging to obtain comparable datasets with consistent acquisition protocols and parameter definitions.

To address this limitation, we have revised the Discussion section (page XXX, lines XXX–XXX) to explicitly highlight these constraints. We also emphasized that, despite being single-center, our dataset encompasses a diverse range of high-risk pregnancy profiles, gestational ages, and monitoring conditions, recorded by multiple clinicians—thus enhancing internal representativeness and model robustness. Future work will focus on establishing multicenter collaborations, performing external validation, and integrating multimodal predictors to further improve model generalizability and clinical applicability.

See lines 846-859 and 872-881 in the revised manuscript.

Comments 2: Feature importance and explainability are not sufficiently analyzed. A clear explanation of which cCTG parameters most influence the prediction would improve clinical trust.

Response 2: To address this important point, a feature importance analysis was performed using the best-performing regressor—the Random Forest model—which achieved the highest predictive accuracy on the validation set. This analysis quantified the relative contribution of each computerized cardiotocography (cCTG) and clinical variable to the prediction of neonatal length of stay, thereby enhancing the interpretability and clinical credibility of the proposed model.

The gestational age at delivery emerged as the dominant predictor, accounting for 56.91% of the model’s decision variance. This result aligns with clinical evidence, as prematurity is a well-established determinant of prolonged neonatal hospitalization. However, this variable explained only slightly more than half of the model’s variance, indicating that multiple additional maternal, fetal, and perinatal factors also contribute meaningfully to hospitalization duration. The gestational age at the time of monitoring ranked second 13.18%, displaying a comparable negative association with hospitalization duration.

Among cCTG-derived features, several parameters had notable contributions to model performance. Fetal movements 4.13% and decelerations 3.62% were both positively associated with hospitalization duration, suggesting that increased fetal activity or abnormal heart rate responses may reflect higher neonatal morbidity and a greater likelihood of prolonged postnatal care. Variability-related indices—including the percentage of STV values <1 bpm 1.91%, average STV 1.31%, and percentage of LTV values <5 bpm 1.27%—also exhibited positive contributions. These findings are consistent with existing evidence that reduced short-term (STV) and long-term variability (LTV) indicate impaired fetal autonomic regulation and suboptimal oxygenation. Additionally, pathological features such as late decelerations further influenced the model outcome, supporting the physiological coherence of its predictive behavior.

Importantly, as noted in the revised manuscript (Introduction, lines 83–86), the interpretation of cCTG-derived parameters—particularly STV and LTV—has been primarily validated for term pregnancies during labor, while their characterization outside the intrapartum period remains less clearly defined. This reinforces the relevance of our feature importance findings, demonstrating that even when assessed outside of labor, variability parameters retain significant predictive value for neonatal outcomes.

Overall, the predominance of gestational maturity and heart rate variability parameters underscores that both developmental stage and autonomic regulation are critical determinants of neonatal adaptation and recovery. These aspects are further emphasized in the Discussion (lines 852–870), where the prognostic value of STV and LTV in reflecting fetal autonomic control and tolerance to hypoxic or metabolic stress is discussed. Future work will extend these explainability analyses to investigate parameter interactions and further enhance model transparency and clinical interpretability.

See lines 808-822 in the revised manuscript.

Comments 3: Comparative baseline models (for example, linear regression or clinical scoring) should be included for benchmarking.

Response 3: We fully agree with the reviewer’s suggestion. In the revised manuscript, a linear regression model with Ridge regularization was implemented and used as a benchmark for performance comparison. The results, now reported in the Results section and summarized in Table 2 and Fig. 3, confirm that the proposed ML regressors significantly outperform the linear baseline in both training and validation sets, supporting the added value of nonlinear ensemble methods.

See section 2.3. Machine learning methods lines 354-3357 and Section 3.1. Performance on Training and Validation Sets lines 465-468, 487-494  in the revised manuscript.

Comments 4: Dataset imbalance and handling of extreme outliers need more discussion.

Response 4: We acknowledge that the distribution of the neonatal length of stay variable is moderately unbalanced, as prolonged hospitalizations are naturally less frequent in clinical practice. This asymmetry reflects the real-world scenario of neonatal care, where most newborns have short or moderate stays, while only a small proportion experience extended hospitalization due to complications.

To mitigate the potential bias introduced by this imbalance, the analysis employed ensemble-based regression models (Random Forest, CatBoost, XGBoost, and LightGBM), which are known to be robust to uneven target distributions and extreme values due to their non-parametric structure and median-based decision mechanisms. Model performance was evaluated using multiple complementary metrics (, RMSE, and MAE), which provide a balanced assessment across the entire outcome range.

Regarding extreme outliers, these cases were retained intentionally, as they represent clinically meaningful situations (e.g., severe intrauterine growth restriction or neonatal distress) rather than measurement errors. Excluding them could have reduced the clinical realism and the model’s generalizability to severe cases.

The potential limitations associated with the unbalanced distribution have now been clearly discussed in the revised manuscript Discussion section ( lines 883-894) .

Future work will aim to address this limitation through prospective data collection and, where feasible, synthetic oversampling approaches for regression problems (e.g., SMOGN — Synthetic Minority Over-sampling for Regression with Gaussian Noise).

See, lines 319-330  and 860-871 in the revised manuscript.

Comments 5: The clinical meaning of a 2-day prediction error should be better interpreted. Does this difference significantly affect decision-making?

Response 5: We thank the reviewer for this valuable observation.

The goal of the paper was to use data-driven approach to predict neonatal length of stay in high-risk pregnancies and not to change the current practice in the obstetrics ward.

The dataset used in our analysis consisted of real-world clinical data, where discharge timing was influenced not only by objective clinical factors but also by subjective and institutional considerations. In many cases, short extensions of hospital stay—typically one to two days—occur due to non-critical reasons such as ensuring adequate feeding and weight gain, short-term phototherapy for mild hyperbilirubinemia, or routine observation prior to discharge. Unforeseen events may also arise during hospitalization, and thus a small prediction margin may even prove beneficial, preventing an overinterpretation of the predicted value.

Therefore, an average prediction error of two days reflects a high level of model precision in estimating a continuous outcome influenced by multiple maternal, fetal, and contextual factors. Clinically, this degree of accuracy is sufficient to support anticipatory planning for neonatal care and resource allocation, without implying any risk of misclassification or inappropriate clinical decisions. Over time, as additional confounding and subjective factors are minimized, the model’s predictive performance is expected to further improve.

The manuscript has been revised accordingly to include this clarification and the detailed interpretation of the two-day prediction error. These additions can be found in the revised version, lines 787–897, where we further explain the clinical acceptability and contextual relevance of the model’s predictive precision.

Comments 6: Figures and tables are numerous; some could be simplified to enhance readability

Response 6: In the revised manuscript, all figures and tables were carefully reviewed to improve clarity and avoid redundancy. Two figures (former Figures 8 and 9) were removed, as their information was already presented in Figure 7 and Table 4, respectively. Additionally, two new figures were added to illustrate the newly implemented validation procedures: the bootstrap-based confidence interval estimation and the permutation-based significance test.

Comments 7: The Discussion section should compare results with more recent neonatal outcome prediction studies and emphasize practical implementation limits.

Response 7: We thank the reviewer for this valuable suggestion. In the revised Discussion section, we have incorporated comparisons with recent studies on neonatal outcome prediction (2022–2024), highlighting both methodological similarities and differences in model performance. Additionally, we expanded the discussion to address the practical limitations of implementing machine learning models in clinical settings, including data heterogeneity, model interpretability, and the need for external validation. These revisions strengthen the contextualization of our results and clarify the scope of real-world applicability. See the revised Discussion section lines 783-786 and 846–881.

Reviewer 2 Report

Comments and Suggestions for Authors

The manuscript demonstrates strong technical execution and an important idea. However, the manuscript must better establish clinical impact, interpretability, and external validation. Incorporating explainable ML analyses, additional clinical covariates, and stronger validation methodology would substantially enhance the study’s translational value.

  1. While Random Forest achieved R² = 0.82 on the test set, the test sample comprises only 61 cases (~9%). Such a small test set may overestimate generalization.
  2. To demonstrate added value, ML performance should be compared against a conventional regression baseline (e.g., multiple linear regression using gestational age + IUGR + HTA + GD + STV). Without this, it is unclear how much ML improves prediction beyond standard statistical models.
  3. Ethical approval is said to be “waived due to retrospective anonymized design.” Nonetheless, the manuscript should include a formal ethics committee approval reference number, even if exemption was granted.
  4. Title – Consider simplifying:
    “Prediction of Neonatal Length of Stay in High-Risk Pregnancies Using Regression-Based Machine Learning on Computerized Cardiotocography Data.”
    Current phrasing (“Computerized CTG Data in High-Risk Pregnancies Using Regression-Based Machine Learning Models”) is somewhat redundant.
  5. Introduction – Although well-referenced, it is disproportionately heavy on prior DL studies (>100 lines) and delays articulation of the study hypothesis. Condense literature background to half its length and foreground the study’s specific aim.
  6. Methods – Provide details on data imputation strategy (if any), software packages and versions (e.g., scikit-learn 1.x, Optuna x.x).
  7. English is clear but occasionally verbose. Minor stylistic adjustments (e.g., “Random Forest achieved the highest generalization performance,” not “showing the strongest generalization”) would improve readability.

Author Response

We thank Reviewer 2 for the constructive feedback and insightful recommendations. The suggestions have been instrumental in refining the clinical focus and enhancing the methodological robustness of the manuscript. The modifications carried out in the revised manuscript are in red.

Comments 1: While Random Forest achieved R² = 0.82 on the test set, the test sample comprises only 61 cases (~9%). Such a small test set may overestimate generalization.

Response 1: We thank the reviewer for this important remark. The full dataset contained 694 records, which were split into 508 training (73.2%), 125 validation (18.0%), and 61 test (8.8%) cases. We acknowledge that a test set of 61 cases provides a limited basis for estimating generalization and may yield an optimistic point estimate of R². To address this limitation, we conducted additional robustness analyses: (i) bootstrap-derived 95% confidence intervals for the test R², and (ii) permutation testing to verify that the observed performance significantly exceeds chance. These complementary procedures provide statistical uncertainty bounds and significance validation for the reported R², reinforcing confidence in the model’s reliability on unseen data. The new results are presented in Section 3.2. Performance on Test Set  (lines 608–645) and Figures 7 and 8 of the revised manuscript. Furthermore, future work will aim to validate the model on independent external cohorts to further confirm generalization.

Comments 2: To demonstrate added value, ML performance should be compared against a conventional regression baseline (e.g., multiple linear regression using gestational age + IUGR + HTA + GD + STV). Without this, it is unclear how much ML improves prediction beyond standard statistical models.

Response 2: Thank you for this valuable comment. To address it, we have added a linear regression baseline (Ridge regularized) to serve as a classical, interpretable benchmark. Its results, presented in Table 2 and Figure 3, were compared with those of the advanced ML regressors (Random Forest, CatBoost, XGBoost, LightGBM), clearly demonstrating superior predictive accuracy and generalization of the ensemble-based models. This addition strengthens the methodological completeness and transparency of the study. See section 2.3. Machine learning methods lines 354-3357 and Section 3.1. Performance on Training and Validation Sets lines 465-468, 487-494  in the revised manuscript.

Comments 3: Ethical approval is said to be “waived due to retrospective anonymized design.” Nonetheless, the manuscript should include a formal ethics committee approval reference number, even if exemption was granted.

Response 3: We are grateful to the reviewer for this helpful observation. We confirm that the study was conducted with formal institutional approval. Specifically, authorization for data access and processing was granted by the management and the Ethics Committee of the Filantropia Clinical Hospital, Bucharest, for the purpose of Dr. Bianca Danciu’s PhD research and any related publications or presentations. The approval was issued under registration number 10528 / 20.09.2022.

Although the study involved retrospective analysis of fully anonymized data, ethical oversight was obtained to ensure compliance with institutional and national regulations regarding patient data confidentiality and research conduct. This information has been added to the revised version of the manuscript. See lines 891-895 in the revised manuscript.

Comments 4: Title – Consider simplifying:
Current phrasing (“Computerized CTG Data in High-Risk Pregnancies Using Regression-Based Machine Learning Models”) is somewhat redundant.

Response 4: We thank the reviewer for this helpful suggestion. We agree that the original title was unnecessarily complex and have revised it accordingly. The new title — “Prediction of Neonatal Length of Stay in High-Risk Pregnancies Using Regression-Based Machine Learning on Computerized Cardiotocography Data” — more clearly reflects the study objectives while maintaining scientific precision and readability. As noted, this change has been implemented in the revised version of the manuscript.

Comments 5: Introduction – Although well-referenced, it is disproportionately heavy on prior DL studies (>100 lines) and delays articulation of the study hypothesis. Condense literature background to half its length and foreground the study’s specific aim.

Response 5: We appreciate the reviewer’s insightful comment. The Introduction section has been substantially revised to improve focus and readability. The background discussion of previous deep learning studies was condensed to approximately half its original length, while preserving the key references necessary for context. The study’s motivation and specific hypothesis are now presented earlier and more explicitly, ensuring a clearer link between the literature review and the research objectives. See the Introduction, lines 115-187 in the revised manuscript.

Comments 6: Methods – Provide details on data imputation strategy (if any), software packages and versions (e.g., scikit-learn 1.x, Optuna x.x).

Response: The Results section has been updated to include details on data handling and software environments. No data imputation was required, as the dataset contained no missing values. Additionally, information on all software packages and versions used in model development (including Python libraries such as scikit-learn and CatBoost/XGBoost/LightGBM stable releases) has been added for full reproducibility. See  the Introduction lines 277–280 and Section3.1. Performance on Training and Validation Sets  530-541 in the revised manuscript.

Comments 7: English is clear but occasionally verbose. Minor stylistic adjustments (e.g., “Random Forest achieved the highest generalization performance,” not “showing the strongest generalization”) would improve readability.

Response 7: We appreciate the reviewer’s helpful linguistic observation. The manuscript has been carefully revised to improve clarity and conciseness throughout. Several sentences were restructured to enhance readability and precision, and minor stylistic and grammatical adjustments were applied consistently to align with academic English standards suitable for MDPI publications.

Round 2

Reviewer 1 Report

Comments and Suggestions for Authors

The authors addressed all my concerns. The paper can be accepted for publication.

Reviewer 2 Report

Comments and Suggestions for Authors

Necessary changes have been made.